palaeontology

biomechanical analysis, digital visualization, fossils, three-dimensional models

**Author for correspondence:**
Stephan Lautenschlager
e-mail: s.lautenschlager@bham.ac.uk

# True colours or red herrings?: colour maps for finite-element analysis in palaeontological studies to enhance interpretation and accessibility

## Stephan Lautenschlager

School of Geography, Earth and Environmental Sciences, University of Birmingham, Birmingham, UK

SL, 0000-0003-3472-814X

Accessibility is a key aspect for the presentation of research data. In palaeontology, new data is routinely obtained with computational techniques, such as finite-element analysis (FEA). FEA is used to calculate stress and deformation in objects when subjected to external forces. Results are displayed using contour plots in which colour information is used to convey the underlying biomechanical data. The *Rainbow* colour map is nearly exclusively used for these contour plots in palaeontological studies. However, numerous studies in other disciplines have shown the *Rainbow* map to be problematic due to uneven colour representation and its inaccessibility for those with colour vision deficiencies. Here, different colour maps were tested for their accuracy in representing values of FEA models. Differences in stress magnitudes ($\Delta S$) and colour values ($\Delta E$) of subsequent points from the FEA models were compared and their correlation was used as a measure of accuracy. The results confirm that the *Rainbow* colour map is not well suited to represent the underlying stress distribution of FEA models with other colour maps showing a higher discriminative power. As the performance of the colour maps varied with tested scenarios/stress types, it is recommended to use different colour maps for specific purposes.

# 1. Introduction

The last two decades have witnessed a surge in the use of computational techniques to study the anatomy and functional

**Figure 1.** Problems of the *Rainbow* colour scheme: (*a*) non-uniform distances between individual colours (adapted from [27]). (*b*) Lack of intuitive perceptual order. (*c*) *Rainbow* colour map as seen without and with colour vision deficiency (i.e. deuteranopia and protanopia type) and in greyscale.

morphology of fossil organisms with the aim of reconstructing their palaeobiology [1–3]. Tools for the biomechanical analysis of fossils, such as finite-element analysis (FEA) [4,5], computational fluid dynamics (CFD) analysis [6,7] and multibody dynamics analysis (MDA) [8], are now routinely applied to investigate the form–function relationships of fossils. Of these, FEA has become a powerful and ubiquitous method to test hypotheses about the functional capabilities of extinct organisms, in particular for species for which no living analogues may exist.

Originally developed as an engineering technique, FEA predicts the deformation in objects with complex geometries and different materials subject to external load forces. Key to the technique is the subdivision (discretization) of the analysed object into numerous, small and geometrically simple elements connected by shared nodes, for which the deformation calculations are subsequently performed. This simplification allows a quick but generally accurate approximation to solve the problem for any given object and the calculation of biologically relevant performance measures, such as stress and strain [5]. Based on the discretization, discrete stress or strain values can be associated with each element and node in an FEA model. For the presentation of the results, these values can be reported quantitatively, for example as model averages or mean values [9,10], values of individually selected elements or sections [11,12], as stress intervals [13], or using a landmark-based approach on the deformed models [14]. However, reporting a large amount of numerical values may not intuitively convey the observed results to the reader. Therefore, it is common practice to present FEA results more qualitatively in the form of contour plots. For these colour-indexed (pseudo- or false-colour) plots, the numerical value of each element in an FEA model is represented by different colours. Such colour coding can be a powerful tool to differentiate and convey information. Although the use of different colour maps does not change the underlying results (e.g. stress and strain magnitudes), they have a substantial impact on the results' legibility and therefore accessibility of the same. This is particularly true for the use of FEA in a comparative context, which aims to identify (subtle) differences between models (e.g. species) [5].

Traditionally, and with very few exceptions [15–17], the colour scheme of choice for FEAs of palaeontological and biological specimens has been (and still is) the classic *Rainbow* colour map. It is based on the colours in the visible light spectrum from blue (usually lower values) via green, yellow and orange to red (usually higher values). It is one of the most common colour schemes for data visualization and the default option in many software toolkits. Despite its ubiquitous use and popularity, a number of studies in the last two decades have identified considerable problems with the *Rainbow* colour map [18–25]: (i) the perceived transitions between the individual colours of the *Rainbow* map are not uniform [25,26], with some colours (i.e. red, green) seemingly taking up a larger part of the colour map (figure 1*a*). This effect can simulate sharp transitions in sequential data, making small variations in the underlying data appear more important [27,28]. Similarly, yellow is the brightest colour in the *Rainbow* colour map. Although it is not at the extreme end of the colour map, it tends to attract the eye more than other colours in the spectrum [25,29,30]. (ii) While ordered from shorter (blue) to longer (red) wavelengths, the *Rainbow* map does not follow any naturally perceived order. This means that in contrast with greyscale or gradient colour maps (which can be arranged from dark to light or vice versa), there is no implicit order to the *Rainbow* colour map [23,26,31], making the comparison between two relative values difficult (figure 1*b*). (iii) Lastly, but importantly,

the *Rainbow* colour map creates considerable accessibility problems for those with colour vision deficiencies (CVDs). Approximately 5–10% of the population may suffer from some form of CVD, such as red-green blindness (Deuteranopia), which renders data represented by the *Rainbow* colour map largely unreadable [32–34]. Furthermore, similar issues arise when results using the *Rainbow* colour map are converted to a greyscale format, such as for example for printing.

Given these inherent problems with the *Rainbow* colour scheme, several disciplines, including oceanography [35], meteorology [36,37] and geosciences [25,38], have started to address this issue and proposed the use of alternative colour schemes. Here, different colour maps are tested and their effectiveness for the visualization of FEA results of palaeontological models is evaluated.

# 2. Material and methods

In order to evaluate their visual effect and accessibility, different colour maps were tested for a variety of FEA models of fossil specimens and different FEA stress measures. In addition to the traditional *Rainbow* map (see also [39]), nine further, established colour maps were selected (version numbers are provided where present): (a) The five sequential colour maps *Batlow (7.0)*, *Inferno*, *Parula*, *Viridis* and *YlGnBu* [25,40–43]. Sequential colour maps vary between two colours ranging from dark to light (or vice versa) and are suitable for ordered data ranging gradually from low to high values (i.e. ratio data with an absolute zero value) [27]. (b) The three diverging colour maps *Cork (7.0)*, *Polar* and *Roma (7.0)* [25,43]. Diverging colour maps range between two contrasting colours at either end separated by a neutral colour in the middle and are suitable for interval data that can have positive and negative values [27]. (c) As a further option, a variant of the classic *Rainbow* colour map known as *Turbo* was included in the analysis. Although *Turbo* similarly consists of a sequence of colours in the visible light spectrum, it has been suggested to represent a perceptually improved rainbow map with a uniform luminance [44,45]. All colour maps are non-proprietary, in some cases, versioned and available/defined via the respective references above. Not all colour maps are readily and equally available by default in all software but can be added in most cases (see also below),

Other colour maps, such as qualitative, categorical or cyclic colour maps, were not tested as these are not appropriate for FEA data. The colour maps tested here were selected following their use and popularity in different applications. However, not all of the colour maps are perceptually uniform (e.g. the difference between two colours as perceived by the human eye is proportional to the numerical distance within the given colour space). *Batlow*, *Cork*, *Inferno*, *Roma*, *Viridis* and *YlGnBu* are all perceptually uniform, whereas *Parula*, *Polar*, *Rainbow* and *Turbo* are not (see also [43]).

All colour maps used for this study consist of 24 individual colour values (definitions (order and HEX colour codes) are available in the electronic supplementary material), and all outputs presented here were generated in Abaqus and model views were saved as image files. Depending on the software, custom colour maps can be created. In this example, all colour maps were created in Abaqus via a command-line script detailing the colour components via HEX codes individually (see the script in electronic supplementary material). Alternatively, new colour maps (so-called spectra in Abaqus) can be created via a tools menu and selecting successive colours via a colour picker. This process will differ for individual software. However, specific pre-designed colour maps can be generated and accessed via online tools, such as https://colorbrewer2.org/#type=sequential&scheme=BuGn&n=3 [40].

Several FEA models of fossil specimens and different skeletal elements were used here to evaluate the perceptual effects of the tested colour maps with different three-dimensional morphologies: (a) a simplified, planar model of the mandible of the sabre-toothed cat *Dinofelis cristata* as used in [46] (figure 2). This model was chosen to represent a geometrically simple morphology as used for FEA models not derived from computed tomography (CT) or surface-based digitization methods [47,48]. For this model, contour plots displaying the distribution of von Mises stress were chosen as an example for ratio data. (b) A three-dimensional model of the mandible of *Thrinaxodon liorhinus* representing a geometrically more complex morphology compared to the model of *Dinofelis* and derived from CT scanning [49]. For the contour plots, tensile (positive) and compressive (negative) absolute stresses were displayed as an example for interval data. (c) A model of the skull of the therizinosaurian dinosaur *Erlikosaurus andrewsi* as used in [50]. In addition to the different contour plots displaying von Mises stress, the models were also displayed as perceived with deuteranopia-type CVDs. For this purpose, the images were converted accordingly using Adobe Photoshop CC 2020. (d) A model of the skull of the capitosaurian temnospondyl *Parotosuchus helgolandicus* [51] representing a dorsoventrally flattened skull morphology. In addition to the

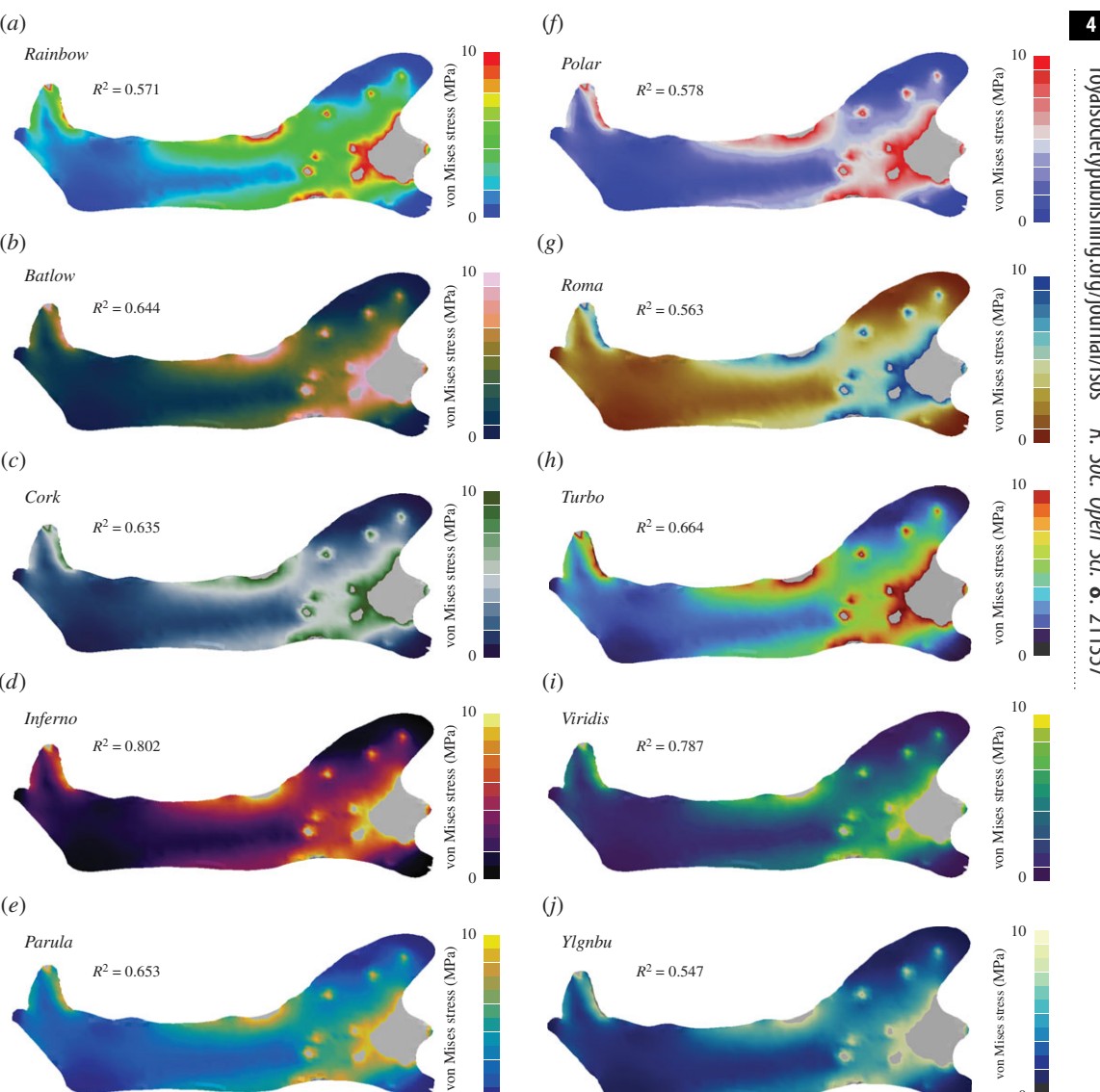

**Figure 2.** Contour plots for different colour maps for Von Mises stress values are shown for the simplified planar mandible model of the sabre-tooth cat *Dinofelis cristata*. In addition to the standard *Rainbow* colour map (*a*), nine further colour maps were tested: *Batlow* (*b*), *Cork* (*c*), *Inferno* (*d*), *Parula* (*e*), *Polar* (*f*), *Roma* (*g*), *Turbo* (*h*), *Viridis* (*i*) and *YlGnBu* (*j*). Grey regions in the contour plots represent stress magnitudes beyond the applied scale limit. $R^2$-values are given for each colour map (see electronic supplementary material for full correlation plots).

different contour plots displaying von Mises stress, the models were also displayed as perceived with a protanopia-type CVD. For this purpose, the images were converted accordingly using Adobe Photoshop CC 2020. (*e*) A model of a dorsal vertebra of the ornithischian dinosaur *Stegosaurus stenops* [47] representing a post-cranial skeletal element. In addition to the different contour plots displaying von Mises stress, the images of the contour plot models were also converted into greyscale using Adobe Photoshop CC (2020) (*Image -> Adjustments -> Black & White* and using the default setting for greyscale mode). (*f*) A model of the manual claw of the therizinosaurian dinosaur *Nothronychus graffami* [52] to illustrate the effect of colour maps against a different background colour. The boundary conditions for these models have no direct effect on the colour map interpretation. Therefore, please refer to the original publications for further details on the boundary conditions of the respective models.

To quantify the discriminative power of the individual colour maps (i.e. relating individual colour values to their respective FEA stress magnitudes), the correlation between the colour maps and stress results of the FEA models was calculated. For consistency across the models, 20 points (i.e. elements)

along with a line across the FEA model were selected that covered the morphology evenly. Stress magnitudes (von Mises and compressive/tensile stresses) were recorded for all sampled points. This approach follows the practice to sample a subset of elements of an FEA model [11,12,50] to quantify its biomechanical properties.

In the next step, the colour values for the sampled points for each tested colour map were recorded as RGB values. Although defining colour as RGB values is a common practice for many (web-based and digital) applications, they were specifically designed for use on monitors and do not reflect human colour perception as the RGB colour space is not uniform [53]. A solution to this problem is using the CIELAB (also known as CIE L*a*b*) colour space [54] which has been designed to be perceptually uniform. Here, the distance between two points defining individual colours is proportional to the perceptual difference between them [55,56]. Therefore, the collected RGB values were converted into CIELAB colour values. The collection of the RGB colours from images of the FEA models and subsequent conversion to CIELAB colour space was done via the convertColor function in R [57] (see electronic supplementary material). For a colour map to represent the underlying data correctly, it must reflect changes in magnitude between two sampled points accordingly. To test this correlation, the absolute difference $\Delta S$ in stress magnitude was calculated for subsequent points sampled for each model (equation (2.1)).

$$\Delta S = \text{stress magnitude of point 1–stress magnitude of point 2.} \tag{2.1}$$

Similarly, the difference in colour value $\Delta E$ was determined for each colour map [53] by calculating the Euclidean distances between two subsequent points (equation (2.2))

$$\Delta E = ((L_1 - L_2)^2 + (a_1 - a_2)^2 + (b_1 - b_2)^2)^{0.5}. \tag{2.2}$$

In a final step, $\Delta E$ and $\Delta S$ were subjected to an ordinary least square regression and the $R^2$-value was obtained as a measure for the discriminative power of the individual colour maps (see electronic supplementary material).

It should be noted that Abaqus applies a shading algorithm when displaying FEA contour plots in that it simulates an artificial light source positioned to the top left of the three-dimensional space. To avoid the effects of artificial shadows on the colour representation, all measurements were performed with the model exposed to the maximum light intensity (usually with models in left lateral or dorsal view).

## 3. Results

Overall, 10 different colour maps were tested for their accuracy to represent the underlying stress magnitudes of finite-element models in the form of contour plots. In addition to the default *Rainbow* map, nine further colour maps were tested and the $R^2$-value was used as a measure for the correlation between stress magnitudes and colour maps.

Across the different colour maps, models, stress types and visual appearances, the $R^2$ values range from nearly no ($R^2 = 0.008$) to strong correlations ($R^2 = 0.967$) (table 1). No single colour map was found to show consistently the strongest correlation for the different test settings, with rather more nuanced variations in representative performance for the different colour maps. It is noteworthy that the *Rainbow* colour map performed worse than most of the other colour maps.

For results in the form of ratio data, such as von Mises stress (figure 2; electronic supplementary material, figure S1), the sequential colour map *Inferno* produced the highest correlation ($R^2 = 0.802$). By contrast, the commonly used *Rainbow* map showed only a weak correlation ($R^2 = 0.571$), and only the colour maps *Roma* ($R^2 = 0.563$) and *YlGnBu* ($R^2 = 0.547$) had a weaker performance. The rainbow variant *Turbo* performed only moderately better than the classic *Rainbow* ($R^2 = 0.664$).

For interval data, such as compressive (i.e. negative) and tensile (i.e. positive) stresses plotted together (figure 3; electronic supplementary material, figure S2), the sequential colour maps *YlGnBu* ($R^2 = 0.967$) and *Parula* ($R^2 = 0.959$), as well as the diverging colour map *Polar* ($R^2 = 0.934$) showed the highest correlation between stress magnitudes and colour representation. The *Rainbow* colour map produced a strong, although not the highest, correlation ($R^2 = 0.887$), whereas *Turbo* performed worst in this scenario but with still a strong correlation ($R^2 = 0.852$).

To test for the discriminative performance of the different colour maps when perceived with a CVD, contour plots were converted to deuteranopia- and protanopia-type appearances (figures 4 and 5;

**Table 1.** $R^2$-values for all tested colour maps stress and visual appearances. Score with the highest value italicized for each test setting.

| | Batlow | Cork | Inferno | Parula | Polar | Rainbow | Roma | Turbo | Viridis | YlGnBu |
|---|---|---|---|---|---|---|---|---|---|---|
| von Mises | 0.644 | 0.635 | 0.802 | 0.653 | 0.578 | 0.571 | 0.563 | 0.664 | 0.787 | 0.547 |
| tensile/compressive | 0.910 | 0.871 | 0.905 | 0.959 | 0.934 | 0.887 | 0.872 | 0.852 | 0.933 | 0.967 |
| deuteranopia | 0.890 | 0.696 | 0.693 | 0.460 | 0.604 | 0.504 | 0.476 | 0.117 | 0.475 | 0.511 |
| protanopia | 0.547 | 0.555 | 0.738 | 0.642 | 0.458 | 0.564 | 0.485 | 0.717 | 0.876 | 0.670 |
| greyscale | 0.476 | 0.440 | 0.578 | 0.360 | 0.348 | 0.086 | 0.098 | 0.008 | 0.306 | 0.500 |

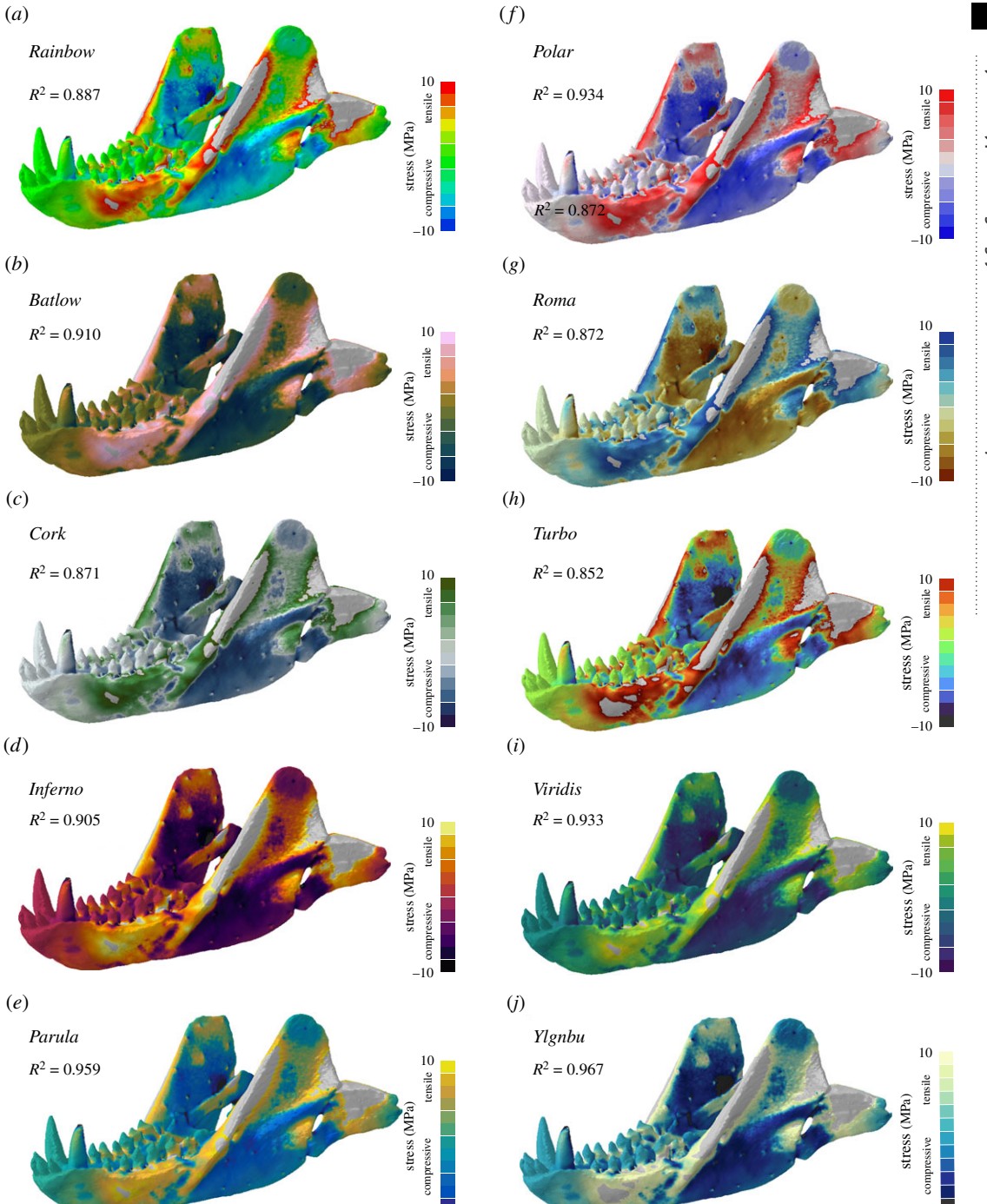

**Figure 3.** Contour plots for different colour maps for compressive and tensile stress values shown for the mandible model of the cynodont *Thrinaxodon liorhinus*. In addition to the standard *Rainbow* colour map (*a*), nine further colour maps were tested: *Batlow* (*b*), *Cork* (*c*), *Inferno* (*d*), *Parula* (*e*), *Polar* (*f*), *Roma* (*g*), *Turbo* (*h*), *Viridis* (*i*) and *YlGnBu* (*j*). Grey regions in the contour plots represent stress magnitudes beyond the applied scale limit. $R^2$-values are given for each colour map (see electronic supplementary material for full correlation plots).

electronic supplementary material, figures S3 and S4). For the deuteranopia type, the sequential colour map *Batlow* ($R^2 = 0.89$) and the diverging colour map *Cork* ($R^2 = 0.696$) were found to represent the stress data the most accurately. Again, the *Rainbow* colour map ($R^2 = 0.504$) was not able to represent the underlying stress results fully, while *Turbo* showed only a very weak correlation ($R^2 = 0.117$). For the protanopia-type contour plots, *Viridis* ($R^2 = 0.876$) and *Inferno* ($R^2 = 0.738$) showed high correlation scores, whereas *Polar* ($R^2 = 0.458$) recorded only a weak correlation.

R. Soc. Open Sci. **8**: 211357

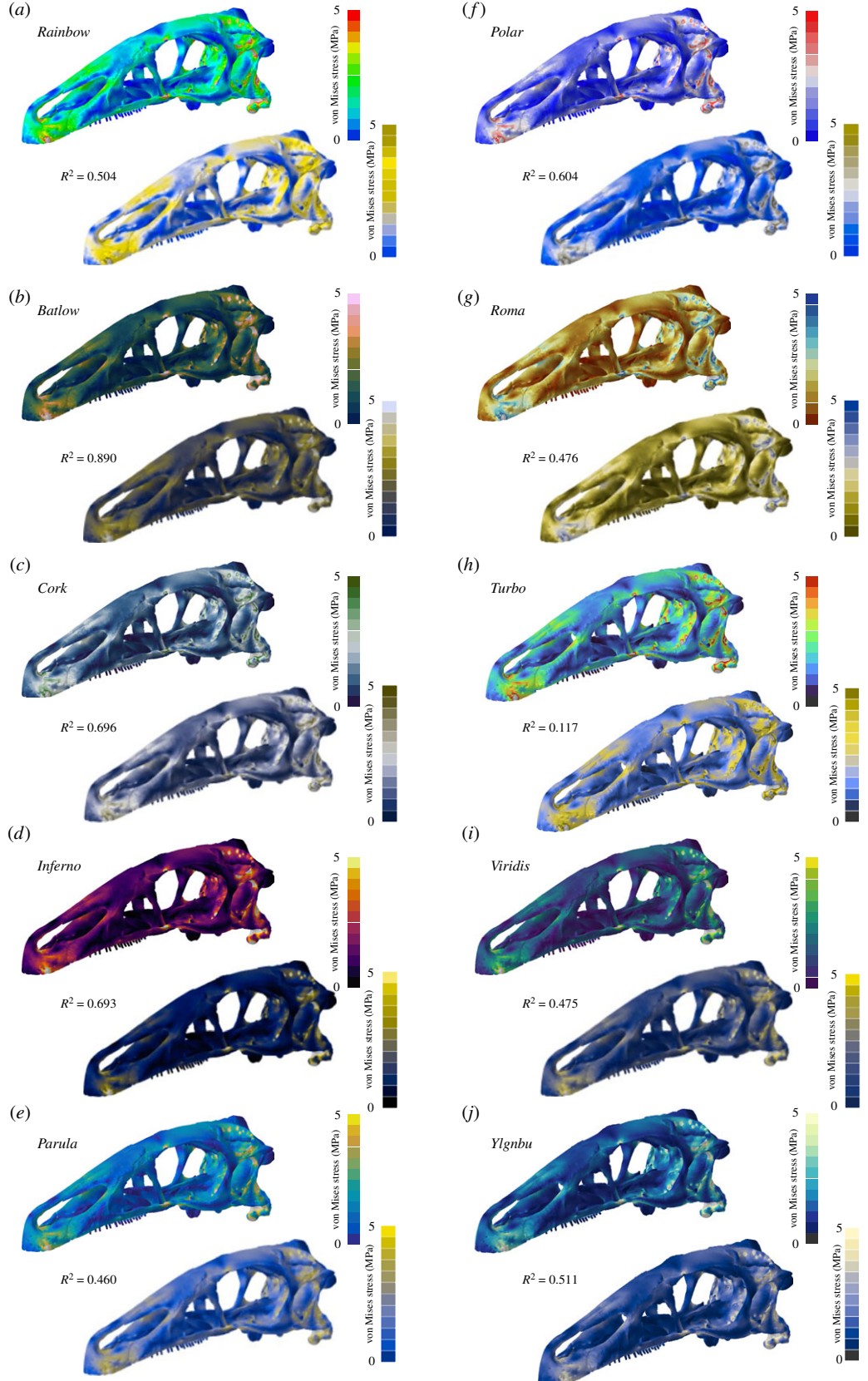

**Figure 4.** Contour plots as seen without and with deuteranopia-type colour vision deficiency for different colour maps. Von Mises stress are values shown for the cranium model of the dinosaur *Erlikosaurus andrewsi*. In addition to the standard *Rainbow* colour map (*a*), nine further colour maps were tested: *Batlow* (*b*), *Cork* (*c*), *Inferno* (*d*), *Parula* (*e*), *Polar* (*f*), *Roma* (*g*), *Turbo* (*h*), *Viridis* (*i*) and *YlGnBu* (*j*). Grey regions in the contour plots represent stress magnitudes beyond the applied scale limit. $R^2$-values are given for each colour map (see electronic supplementary material for full correlation plots).

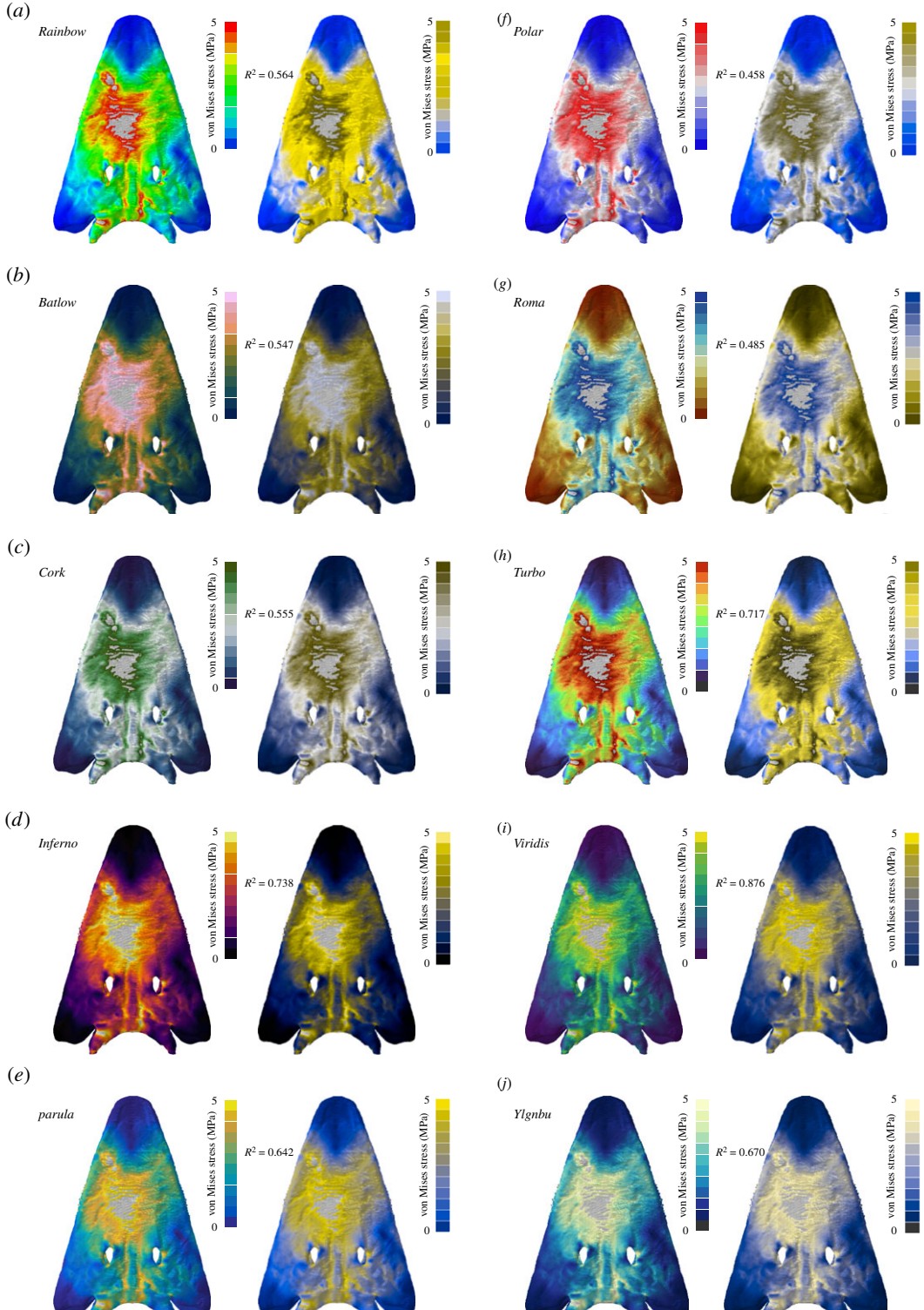

**Figure 5.** Contour plots as seen without and with protanopia-type colour vision deficiency for different colour maps. Von Mises stress values are shown for the cranium model of the capitosaurian temnospondyl *Parotosuchus helgolandicus*. In addition to the standard *Rainbow* colour map (*a*), nine further colour maps were tested: *Batlow* (*b*), *Cork* (*c*), *Inferno* (*d*), *Parula* (*e*), *Polar* (*f*), *Roma* (*g*), *Turbo* (*h*), *Viridis* (*i*) and *YlGnBu* (*j*). Grey regions in the contour plots represent stress magnitudes beyond the applied scale limit. $R^2$-values are given for each colour map (see electronic supplementary material for full correlation plots).

In a final analysis, contour plots were converted to greyscale and the discriminative performance of the colour maps was tested (figure 6; electronic supplementary material, figure S5). In this scenario, all colour maps produced only a moderate to no correlation ($0.578 < R^2 < 0.008$).

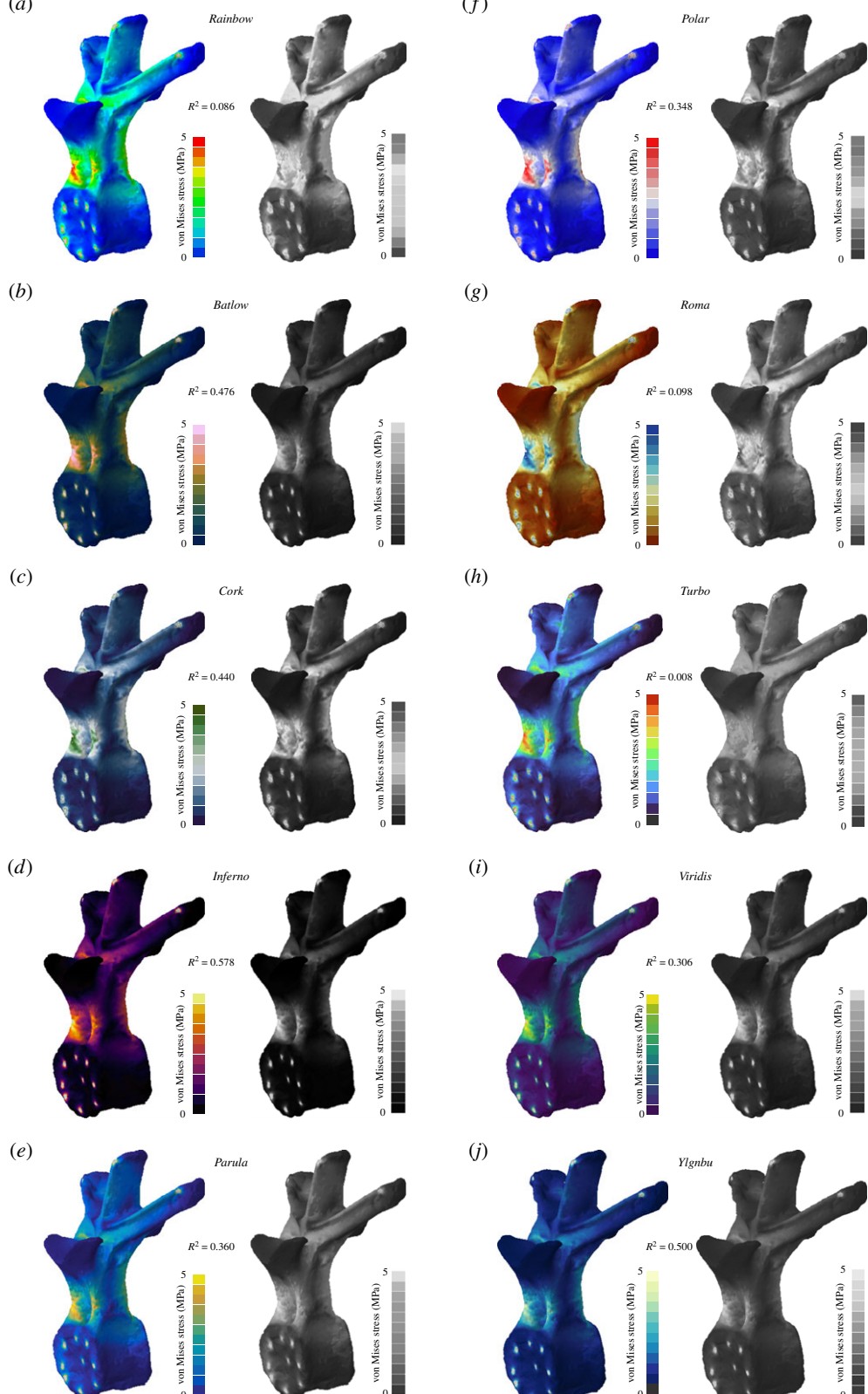

**Figure 6.** Contour plots as seen in full colour and greyscale for different colour maps. Von Mises stress values are shown for a vertebra of the ornithischian dinosaur *Stegosaurus stenops*. In addition to the standard *Rainbow* colour map (*a*), nine further colour maps were tested: *Batlow* (*b*), *Cork* (*c*), *Inferno* (*d*), *Parula* (*e*), *Polar* (*f*), *Roma* (*g*), *Turbo* (*h*), *Viridis* (*i*) and *YlGnBu* (*j*). Grey regions in the contour plots represent stress magnitudes beyond the applied scale limit. $R^2$-values are given for each colour map (see electronic supplementary material for full correlation plots).

The best performance was found for *Inferno* ($R^2 = 0.578$); *Rainbow, Roma* and *Turbo* showed the least correlation ($R^2 < 0.098$).

## 4. Discussion

The *Rainbow* colour map has been a ubiquitous tool in data visualization for decades [25,37]. Engineering techniques such as FEA, which has increasingly been used in palaeontological studies over the last 20 years, are no exception. Results from FEAs are routinely visualized in the form of contour plots using the *Rainbow* colour map. Contour plots typically display von Mises stress, a common measure to evaluate the stability of a model under loading conditions. However, as shown by the results from this study, the *Rainbow* colour map correlates only poorly with the underlying von Mises stress data (table 1 and figure 2), and its discriminative power is equally poor when perceived with different types of CVDs (figures 4 and 5). This should not come as a surprise as the *Rainbow* colour map has been considered problematic and misleading in other disciplines [18,19,22,23–25].

Other colour maps tested here performed considerably better. However, no one colour map was found to be optimally suited for all types of stress and visual perception. For interval-type stresses, such as compressive and tensile stresses plotted on the same model, the *Rainbow* colour map showed a high correlation (table 1, figure 3) similar to or even better than the diverging colour maps in this study. Interestingly, diverging colour maps did not necessarily perform better for interval data, whereas sequential colour maps were not always found to show the best correlation for ratio data (i.e. von Mises stress). *Inferno, Batlow* and *Parula* generally showed the highest discriminative power, but not consistently so (table 1). It is noteworthy that differences in the performance were recorded when colour maps were tested in CVD settings. The same colour maps (*Inferno, Batlow* and *Parula*, and to a lesser degree *Polar* and *Viridis*) represented the underlying stress values reasonably well despite the reduced colour information. However, this means that a single colour map cannot be used as a silver bullet to perform equally well under all conditions. Similar to the alternative text describing figures, the second set of contour plots with a different colour map could be provided in the electronic supplementary material accessible to those with CVD. More generally, the use of specific colour maps may have to be decided on a case to case basis using custom-made or existing colour maps (see, for example, [38] for available colour maps). For interval-type data, other considerations than the discriminative power (expressed as the $R^2$ value here) may need to be considered. For such data, the central zero value can be an important identifier of stress-free regions in the model, which can be recognized more easily when diverging colour maps are used.

In this context, it should be noted that the correlation analysis used here to discriminate stress/colour changes is not perfect. Human colour perception is not uniform, often subjective and dependent on other factors such as age and individual variation and as such does not correspond to Euclidean distances in colour space [58]. The CIELAB colour space is an attempt to replicate human colour differentiation. As the correlation analysis only considers absolute changes along with a trajectory, the analysis may not record the exact correlation when non-monotonic changes on the stress scale are associated with changes in different directions in the CIELAB space. However, this is less likely to be a problem for the perceptually based colour maps.

The choice of an appropriate colour map may further depend on the nature of the results of an FEA. Models spanning a wide range of stress magnitudes, but with an uneven distribution of values will be biased towards certain regions of the colour map. This situation could result in a lower resolution of stress (and thereby colour) values towards the lower end of the colour map to encompass the full range of stress magnitudes present. Sequential colour maps will be a better option in such cases as their colour gradient is expressed along with the whole range of the colour map in comparison to divergent colour maps.

It is important to note that CVD is only one form of visual impairment and of course further improvements for accessibility should be aimed for when considering the presentation of results from FEA (and other analyses more broadly). The Web Content Accessibility Guidelines (WCAG) [59] provide further recommendations to improve accessibility, including appropriate contrast ratios between colours to allow their distinction. For example, *Cork*, *Polar* and *Roma* have high contrast ratios, whereas *Inferno* and *Viridis* have poorer contrast ratios. These ratios are, of course, lower if subsequent colours along with the gradient were to be tested, not just extreme and mid-point values. However, this goes to show that not only the uniform sequence of colours but also the contrast between them plays a role in making contour plots accessible.

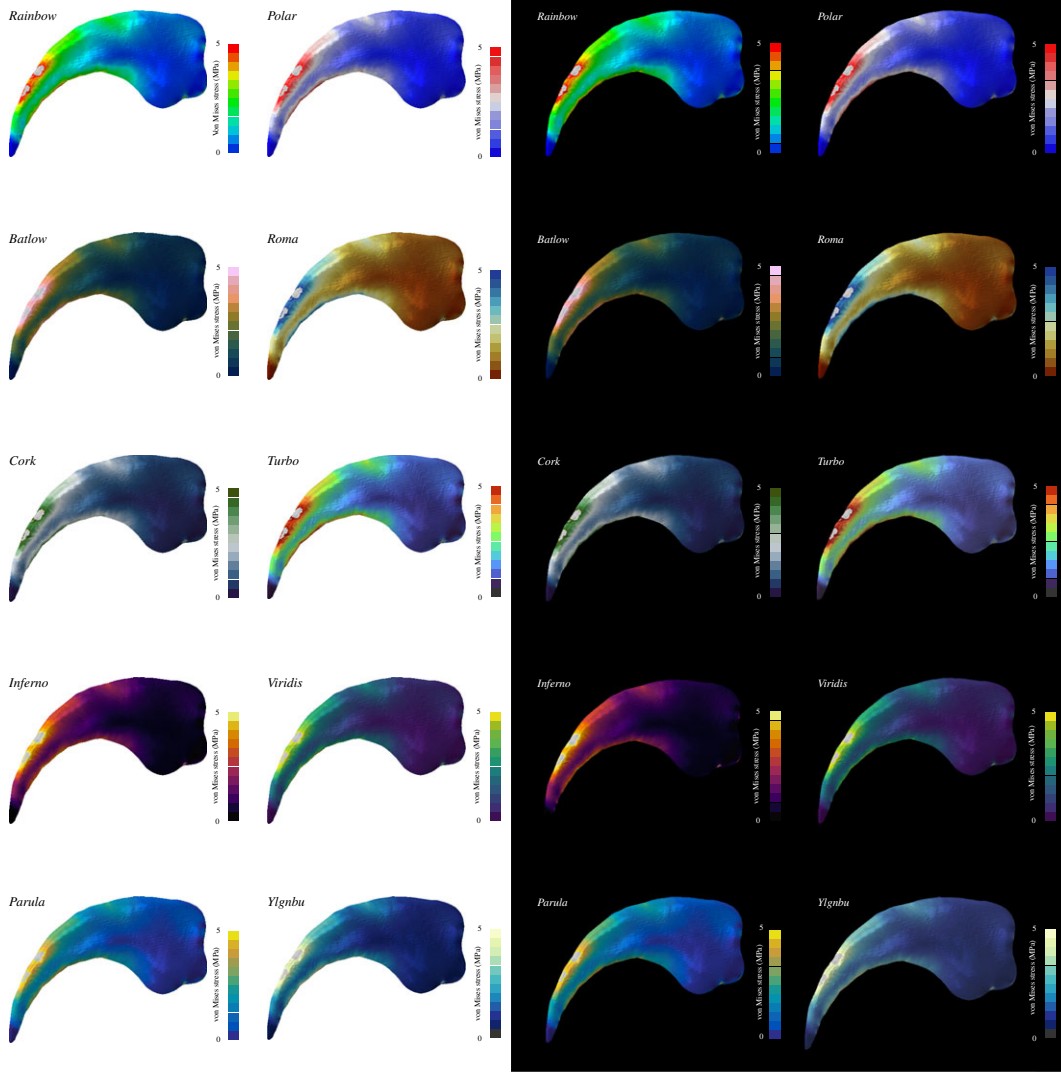

**Figure 7.** Contour plots are depicted in the context of different background colours for the same model and colour maps. von Mises stress values are shown for a manual claw of the therizinosaurian dinosaur *Nothronychus graffami*. Grey regions in the contour plots represent stress magnitudes beyond the applied scale limit.

This situation is further complicated in that FEA contour plots cannot be regarded in isolation but need to be considered in the context of background and environmental settings. In the simplest of cases, this could mean that the choice of background colour can influence the readability of the contour plots [25,60,61] (figure 7). Especially colour maps with a large amount of dark components can become invisible against a black background. Colourmaps with strong contrast and luminosity (e.g. *Parula*, *Polar*) can work well in such a case. For the presentation on a white background, colour maps with a decreasing chroma (=colour intensity), such as *Batlow, Inferno* and, in particular, *YlGnBu* are more appropriate to convey the results [25,61,62].

Within a digital, three-dimensional environment pseudo-colouring creates a further difficulty as the choice of colours interact with the shading and perception of spatial cues [24]. Properties such as the number, direction and intensity of light sources, specularity (i.e. reflectiveness of a surface) and other settings can have an impact on the appearance of colour maps as well. Further, different devices and display technologies display colours variably and for consistent perception colour calibration would have to be performed first.

Most FEA software allows turning off shading effects. However, this could possibly result in a reduced perception of the model morphology, especially for flattened surfaces with low topography (e.g. figure 5). Although not tested here, it should further be taken into consideration that different FEA software packages may use slightly different variations/colour definitions of the *Rainbow* colour map, further exacerbating comparisons between outputs from different software.

The eye-catching quality of the *Rainbow* colour map with its high luminance and contrast is likely the reason for its continued prevalence despite its problems with data distortion. Different reasons have been discussed in the past [24,25] for why the *Rainbow* colour map is still the visualization tool of choice for many studies and applications. For finite-element models, this has likely historic reasons and it is the default colour map in most software. Furthermore, the colour distribution of the *Rainbow* map has a very strong signalling function and communicative power: cold colours (i.e. blue) are associated with no or low stresses, whereas warm colours (i.e. yellow, red) indicate high-stress magnitudes. For von Mises stress, high magnitudes indicate possible material failure and an association with a colour such as red which is commonly used to convey danger is intuitive [63]. However, this concept can also be conveyed with other colour maps such as *Inferno*.

A recent study has used a variety of colour maps to display the results from FEAs [64]. Similarly, for palaeontological studies using other engineering tools, different colour maps have started to appear in publications. CFD, an engineering technique to simulate fluid flow within or around objects, uses a similar approach to FEA to represent data with pseudo-colour plots [7]. Although the *Rainbow* colour map is routinely used to visualize CFD results, different colour maps have been used recently in some studies [65].

## 5. Conclusion

Results from this study demonstrate that the *Rainbow* colour map is not well suited to represent the underlying stress distribution of FEA models. Although most of the other colour maps tested here showed a higher discriminative power, no single colour map was found to perform consistently well throughout all scenarios and for all stress types. It is therefore recommended that different colour maps without data distortion are used to present results. This could mean using different colour maps for ratio (e.g. von Mises stress) and interval data (e.g. compressive and tensile stresses). Alternatively, the second set of contour plots with a different colour map could be provided in the electronic supplementary material to increase accessibility.

The perception of colour is highly dependent on multiple factors, including display devices, colour standards for display and printing, and differences in the human visual apparatus. A variety of different colour maps displayed on different models of palaeontological specimens have been presented here. It is hoped that the reader will use these examples alongside the quantitative evaluation as guidance for their applications and studies. However, the tested colour maps in this study are far from exhaustive and a variety of tools exist to access pre-designed colourmaps (see [25], Box 2).

Ethics. No ethical issues arose in the course of this study.

Data accessibility. Raw measurements and code are included in the electronic supplementary material. FEA results files are available here: https://figshare.com/articles/dataset/FEA_models_from_True_colours_or_red_herrings_-_Colour_maps_for_finite_element_analysis_in_palaeontological_studies_and_how_they_can_enhance_interpretation_and_accessibility_/14905104. The data are provided in the electronic supplementary material [66].

Competing interests. I have no competing interests.

Funding. No funding source to report.

Acknowledgements. Emma Dunne and Sarah Greene kindly provided feedback on the draft manuscript. Fabio Crameri, Melisa Morales-Garcia, Eric Snively, Claus Wilke and Achim Zeileis are thanked for constructive comments and helpful suggestions improving earlier versions of the manuscript.

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
