## [Peer Review File · Royal Society Open Science]

Review History

RSOS-211357.R0 (Original submission)

Review form: Reviewer 1

Is the manuscript scientifically sound in its present form?

Yes

Are the interpretations and conclusions justified by the results?

Yes

Is the language acceptable?

Yes

Do you have any ethical concerns with this paper?

No

Have you any concerns about statistical analyses in this paper?

No

Recommendation?

Accept with minor revision (please list in comments)

Comments to the Author(s)

This is an excellent study and very relevant now. The range of example models and colours is great to give an idea of what can work for future publications. The quantitative results are particularly helpful, because it gives some guidance on the best colour plot for the type of data, rather than a subjective preference based on appearance.

At the start, I expected to change my mind about the rainbow colour plot and want to use something different, but since you conclude that there is no single colour plot that solves “all” issues, I feel like the rainbow plot is still the one that will be chosen in future. Especially because it’s also the default plot from software and is easy to produce.

I particularly like the recommendation though to add a second set of contour plots (or more) with a different colour map in the supplementary material. This could really help those with CVD. I think adding multiple colour plots in the main article may be confusing, and most people are used to the rainbow colour plot.

One thing I’d like clearer is how to apply the colour plots; I found this confusing or unclear and if I was to do this myself, I wouldn’t know how. So, if this can be made clearer in the Methods or supplemental material, that would be great. I found the supplementary figures in the included files, but I could not find the supplementary material that was referred to on pages 4 and 6 in the Methods. If a step-by-step guide is included in the supplementary information, that would be good. The thing I found most confusing is you say the outputs were generated in Abaqus and then converted using convertColor function in R? Were the outputs saved as images and converted from the default rainbow, or were the models exported and the colour was resampled in R? I would love to know how to do it.

Otherwise, very happy to recommend this for publication.

Review form: Reviewer 2

Is the manuscript scientifically sound in its present form?

Yes

Are the interpretations and conclusions justified by the results?

Yes

Is the language acceptable?

Yes

Do you have any ethical concerns with this paper?

No

Have you any concerns about statistical analyses in this paper?

No

Recommendation?

Accept with minor revision (please list in comments)

Comments to the Author(s)

The manuscript has intriguing results for legible visualization that I hope to put into practice soon (see Appendix A). Among minor improvements to the language, I suggest that the authors spell out acronyms more often, and be more specific for lines 217-218.

More substantive but still minor improvements will be:

1. To explain more clearly the commercial propriety or openness of various color schemes, and how generally they're available amongst relevant simulation and visualization programs.
2. To explain how well the mathematical discrimination between colors matches perceptual discrimination.

Decision letter (RSOS-211357.R0)

Dear Dr Lautenschlager

On behalf of the Editors, we are pleased to inform you that your Manuscript RSOS-211357 "True colours or red herrings? - colour maps for finite element analysis in palaeontological studies to enhance interpretation and accessibility" has been accepted for publication in Royal Society Open Science subject to minor revision in accordance with the referees' reports. Please find the referees' comments along with any feedback from the Editors below my signature.

Please submit your revised manuscript and required files (see below) no later than 7 days from today's (ie 12-Oct-2021) date. Note: the ScholarOne system will 'lock' if submission of the revision is attempted 7 or more days after the deadline. If you do not think you will be able to meet this deadline please contact the editorial office immediately.

on behalf of Dr Jennifer Botha (Associate Editor) and Peter Haynes (Subject Editor)
openscience@royalsociety.org

Associate Editor Comments to Author (Dr Jennifer Botha):
Comments to the Author:

This paper has been reviewed by two researchers who are both enthusiastic about the results of the study. They both suggest only minor revision. I do not think it will take much time to incorporate the extra explanations that have been asked for, thus I recommend accept with minor revision.

Reviewer comments to Author:

Reviewer: 1

Comments to the Author(s)

This is an excellent study and very relevant now. The range of example models and colours is great to give an idea of what can work for future publications. The quantitative results are particularly helpful, because it gives some guidance on the best colour plot for the type of data, rather than a subjective preference based on appearance.

At the start, I expected to change my mind about the rainbow colour plot and want to use something different, but since you conclude that there is no single colour plot that solves “all” issues, I feel like the rainbow plot is still the one that will be chosen in future. Especially because it’s also the default plot from software and is easy to produce.

I particularly like the recommendation though to add a second set of contour plots (or more) with a different colour map in the supplementary material. This could really help those with CVD. I think adding multiple colour plots in the main article may be confusing, and most people are used to the rainbow colour plot.

One thing I’d like clearer is how to apply the colour plots; I found this confusing or unclear and if I was to do this myself, I wouldn’t know how. So, if this can be made clearer in the Methods or supplemental material, that would be great. I found the supplementary figures in the included files, but I could not find the supplementary material that was referred to on pages 4 and 6 in the Methods. If a step-by-step guide is included in the supplementary information, that would be good. The thing I found most confusing is you say the outputs were generated in Abaqus and then converted using convertColor function in R? Were the outputs saved as images and converted from the default rainbow, or were the models exported and the colour was resampled in R? I would love to know how to do it.

Otherwise, very happy to recommend this for publication.

Reviewer: 2

Comments to the Author(s)

The manuscript has intriguing results for legible visualization that I hope to put into practice soon. Among minor improvements to the language, I suggest that the authors spell out acronyms more often, and be more specific for lines 217-218.

More substantive but still minor improvements will be:

1. To explain more clearly the commercial propriety or openness of various color schemes, and how generally they're available amongst relevant simulation and visualization programs.
2. To explain how well the mathematical discrimination between colors matches perceptual discrimination.

Attached PDF: RSOS-211357_Proof_hi.pdf

===PREPARING YOUR MANUSCRIPT===

one version identifying all the changes that have been made (for instance, in coloured highlight, in bold text, or tracked changes);
 a 'clean' version of the new manuscript that incorporates the changes made, but does not highlight them. This version will be used for typesetting.

===PREPARING YOUR REVISION IN SCHOLARONE===

- Any electronic supplementary material (ESM).
- If you are requesting a discretionary waiver for the article processing charge, the waiver form must be included at this step.
- If you are providing image files for potential cover images, please upload these at this step, and inform the editorial office you have done so. You must hold the copyright to any image provided.
- A copy of your point-by-point response to referees and Editors. This will expedite the preparation of your proof.

- Ensure that your data access statement meets the requirements at <https://royalsociety.org/journals/authors/author-guidelines/#data>. You should ensure that you cite the dataset in your reference list. If you have deposited data etc in the Dryad repository, please only include the 'For publication' link at this stage. You should remove the 'For review' link.
- If you are requesting an article processing charge waiver, you must select the relevant waiver option (if requesting a discretionary waiver, the form should have been uploaded at Step 3 'File upload' above).
- If you have uploaded ESM files, please ensure you follow the guidance at <https://royalsociety.org/journals/authors/author-guidelines/#supplementary-material> to include a suitable title and informative caption. An example of appropriate titling and captioning may be found at https://figshare.com/articles/Table_S2_from_Is_there_a_trade-off_between_peak_performance_and_performance_breadth_across_temperatures_for_aerobic_scope_in_teleost_fishes_/3843624.

Author's Response to Decision Letter for (RSOS-211357.R0)

See Appendix B.

Decision letter (RSOS-211357.R1)

Dear Dr Lautenschlager,

I am pleased to inform you that your manuscript entitled "True colours or red herrings? - colour maps for finite element analysis in palaeontological studies to enhance interpretation and accessibility" is now accepted for publication in Royal Society Open Science.

If you have not already done so, please ensure that you send to the editorial office an editable version of your accepted manuscript, and individual files for each figure and table included in

your manuscript. You can send these in a zip folder if more convenient. Failure to provide these files may delay the processing of your proof.

on behalf of Dr Jennifer Botha (Associate Editor) and Peter Haynes (Subject Editor)
openscience@royalsociety.org

Appendix A

ROYAL SOCIETY OPEN SCIENCE

True colours or red herrings? - colour maps for finite element analysis in palaeontological studies to enhance interpretation and accessibility

Journal:	Royal Society Open Science
Manuscript ID	RSOS-211357
Article Type:	Research
Date Submitted by the Author:	21-Aug-2021
Complete List of Authors:	Lautenschlager, Stephan; University of Birmingham,
Subject:	Palaeontology < EARTH SCIENCES
Keywords:	biomechanical analysis, digital visualisation, fossils, 3D models
Subject Category:	Earth and Environmental Science

Author-supplied statements

Relevant information will appear here if provided.

Ethics

Does your article include research that required ethical approval or permits?:

This article does not present research with ethical considerations

Statement (if applicable):

CUST_IF_YES_ETHICS :No data available.

Data

It is a condition of publication that data, code and materials supporting your paper are made publicly available. Does your paper present new data?:

Yes

Statement (if applicable):

Raw measurements and code are included in the supplementary information. FEA results files are available here (and may be transferred to a Dryad repository upon acceptance if necessary):

https://figshare.com/articles/dataset/FEA_models_from_True_colours_or_red_herrings_-_Colour_maps_for_finite_element_analysis_in_palaеontological_studies_and_how_they_can_enhance_interpretation_and_accessibility_/14905104

Conflict of interest

I/We declare we have no competing interests

Statement (if applicable):

CUST_STATE_CONFLICT :No data available.

Authors' contributions

I am the only author on this paper

Statement (if applicable):

CUST_AUTHOR_CONTRIBUTIONS_TEXT :No data available.

True colours or red herrings? - colour maps for finite element analysis in palaeontological studies to enhance interpretation and accessibility

Stephan Lautenschlager

School of Geography, Earth and Environmental Sciences, University of Birmingham, Birmingham, UK

Keywords: biomechanical analysis, digital visualisation, fossils, 3D models

1. Summary

Accessibility is a key aspect for the presentation of research data. In palaeontological sciences, new data on the palaeobiology of extinct organisms is routinely obtained with computational techniques, such as finite element analysis (FEA). FEA is used to calculate stress and deformation in objects such as the skulls or limb bones when subjected to external load forces. Results are displayed using false-colour contour plots in which colour information is used to convey the underlying biomechanical data. The *Rainbow* colour map is nearly exclusively used to present these contour plots in palaeontological studies using FEA. However, numerous studies in other disciplines have shown the *Rainbow* colour map to be problematic due to uneven colour representation and its inaccessibility for those with colour-vision deficiencies.

Here, ten different colour maps were tested for their accuracy in representing the underlying stress values of FEA models. Differences in stress magnitudes (ΔS) and colour values (ΔE) of subsequent points taken from the FEA models were compared and their correlation was used as a measure of the accuracy.

The results demonstrate that the *Rainbow* colour map is not well suited to represent the underlying stress distribution of FEA models. Most of the other colour maps tested here showed a higher discriminative power. However, the performance of the different colour maps varied with the different tested scenarios and stress types. It is therefore recommended to use different colour maps for specific stress types.

2. Introduction

[revised manuscript text omitted]

**Ethical Statement**

No ethical issues arose in the course of this study.

**Funding Statement**

No funding source to report.

**Data Accessibility**

Raw measurements and code are included in the supplementary information. FEA results files are available here (and may
be transferred to a Dryad repository upon acceptance if necessary):

[https://figshare.com/articles/dataset/FEA_models_from_True_colours_or_red_herrings_-
_Colour_maps_for_finite_element_analysis_in_palaeontological_studies_and_how_they_can_enhance_interpretation_and
_accessibility_/14905104](https://figshare.com/articles/dataset/FEA_models_from_True_colours_or_red_herrings_-_Colour_maps_for_finite_element_analysis_in_palaeontological_studies_and_how_they_can_enhance_interpretation_and_accessibility_/14905104)

**Competing Interests**

I have no competing interests

**References**

- 1. Cunningham J. A., Rahman I. A., Lautenschlager S., Rayfield E. J., Donoghue P. C. 2014. A virtual world of
paleontology. *Trends in Ecology & Evolution* 29(6): 347-357.
- 2. Sutton M., Rahman I., Garwood R. 2014. *Techniques for virtual palaeontology*. John Wiley Sons.
- 3. Lautenschlager S. 2016. Reconstructing the past: methods and techniques for the digital restoration of fossils.
*Royal Society Open Science* 3(10): 160342.
- 4. Rayfield E. J. 2007. Finite element analysis and understanding the biomechanics and evolution of living and
fossil organisms. *Annual Reviews of Earth and Planetary Sciences* 35 541-576.

5. Bright J. A. 2014. A review of paleontological finite element models and their validity. *Journal of Paleontology* 88(4): 760-769.
6. Bourke J. M., Ruger Porter W. M., Ridgely R. C., Lyson T. R., Schachner E. R., Bell P. R., Witmer L. M. 2014. Breathing life into dinosaurs: tackling challenges of soft-tissue restoration and nasal airflow in extinct species. *The Anatomical Record* 297(11): 2148-2186.
7. Rahman I. A. 2017. Computational fluid dynamics as a tool for testing functional and ecological hypotheses in fossil taxa. *Palaeontology* 60(4): 451-459.
8. Lautenschlager S. 2020. Multibody dynamics analysis (MDA) as a numerical modelling tool to reconstruct the function and palaeobiology of extinct organisms. *Palaeontology* 63(5): 703-715.
9. Farke A. A. 2008. Frontal sinuses and head-butting in goats: a finite element analysis. *Journal of Experimental Biology* 211(19): 3085-3094.
10. Lautenschlager S. 2017. Functional niche partitioning in Therizinosauria provides new insights into the evolution of theropod herbivory. *Palaeontology* 60(3): 375-387.
11. Piras, P., Sansalone, G., Teresi, L., Moscato, M., Profico, A., Eng, R., Cox, T.C., Loy, A., Colangelo, P. and Kotsakis, T. 2015. Digging adaptation in insectivorous subterranean eutherians. The enigma of *Mesoscolops montanensis* unveiled by geometric morphometrics and finite element analysis. *Journal of Morphology* 276(10) 1157-1171.
12. Attard M. R., Wilson L. A., Worthy T. H., Scofield P., Johnston P., Parr W. C., Wroe S. 2016. Moa diet fits the bill: virtual reconstruction incorporating mummified remains and prediction of biomechanical performance in avian giants. *Proceedings of the Royal Society B: Biological Sciences* 283(1822): 20152043.
13. Marcé-Nogué J., De Esteban-Trivigno S., Püschel T. A., Fortuny J. 2017. The intervals method: a new approach to analyse finite element outputs using multivariate statistics. *PeerJ* 5 e3793.
14. Lautenschlager S., Brassey C. A., Button D. J., Barrett P. M. 2016. Decoupled form and function in disparate herbivorous dinosaur clades. *Scientific Reports* 6(1): 1-10.
15. Snively E., Russell A. 2002. The tyrannosaurid metatarsus: bone strain and inferred ligament function. *Senckenbergiana Lethaea* 82(1): 35.
16. Kleinteich T., Maddin H. C., Herzen J., Beckmann F., Summers A. P. 2012. Is solid always best? Cranial performance in solid and fenestrated caecilian skulls. *Journal of Experimental Biology* 215(5): 833-844.
17. van der Meijden A., Kleinteich T., Coelho P. 2012. Packing a pinch: functional implications of chela shapes in scorpions using finite element analysis. *Journal of Anatomy* 220(5): 423-434
18. Herman G. T., Levkowitz H. 1992. Color scales for image data. *Computer Graphics and Applications* 12(1): 72-80.
19. Rogowitz B. E., Treinish L. A., Bryson S. 1996. How not to lie with visualization. *Computers in Physics* 10(3): 268-273.
20. Brewer C. A. 1997. Spectral schemes: Controversial color use on maps. *Cartography and Geographic Information Systems* 24(4): 203-220.
21. Rogowitz B. E., Treinish L. A. 1998. Data visualization: the end of the rainbow. *IEEE Spectrum* 35(12): 52-59.
22. Light A., Bartlein P. J. 2004. The end of the rainbow? Color schemes for improved data graphics. *Eos Transactions American Geophysical Union* 85(40): 385-391.
23. Borland D., Taylor R. M. 2007. Rainbow color map (still) considered harmful. *IEEE Computer Graphics and Applications* 27(2): 14-17.
24. Moreland K. 2016. Why we use bad color maps and what you can do about it. *Electronic Imaging* 2016(16): 1-6.
25. Crameri F., Shephard G. E., & Heron P. J. 2020. The misuse of colour in science communication. *Nature Communications* 11(1): 1-10.

26. Wong B, 2010. Points of view: Color coding. *Nature Methods* 7(8): 573–573.
27. Hattab G., Rhyne T. M., Heider D. 2020. Ten simple rules to colorize biological data visualization. *PLoS Computational Biology* 16(10): e1008259
28. Rhyne T. M. 2017. Applying color theory to digital media and visualization. In *Proceedings of the 2017 CHI Conference Extended Abstracts on Human Factors in Computing Systems* 1264-1267.
29. Alexander K. R., Shansky M. S. 1976. Influence of hue, value, and chroma on the perceived heaviness of colors. *Perception & Psychophysics* 19(1): 72-74.
30. Wolfe J. M., Horowitz T. S. 2017. Five factors that guide attention in visual search. *Nature Human Behaviour* 1(3): 1-8.
31. Ware C. 2019. *Information visualization: perception for design*. Morgan Kaufmann.
32. Sharpe L. T., Stockman A., Jägle H., Nathans J. 1999. Opsin genes cone photopigments color vision and color blindness. *Color vision: From genes to perception* 3-51.
33. Simunovic M. P. 2010. Colour vision deficiency. *Eye* 24(5): 747-755.
34. Wong B. 2011. Color blindness. *Nature Methods* 8(6): 441-442.
35. Samsel F., Petersen M., Geld T., Abram G., Wendelberger J., Ahrens J. 2015. Colormaps that improve perception of high-resolution ocean data. In *Proceedings of the 33rd Annual ACM Conference Extended Abstracts on Human Factors in Computing Systems* 703-710.
36. Sherman-Morris K., Antonelli K. B., Williams C. C. 2015. Measuring the effectiveness of the graphical communication of hurricane storm surge threat. *Weather Climate and Society* 7(1): 69-82.
37. Stauffer R., Mayr G. J., Dabernig M., Zeileis A. 2015. Somewhere over the rainbow: How to make effective use of colors in meteorological visualizations. *Bulletin of the American Meteorological Society* 96(2): 203-216.
38. Crameri F. 2018. Geodynamic diagnostics scientific visualisation and StagLab 3.0. *Geoscientific Model Development* 11(6): 2541-2562.
39. Quinan P. S., Padilla L. M., Creem-Regehr S. H., Meyer M. 2019. Examining implicit discretization in spectral schemes. *Computer Graphics Forum* 38(3): 363-374.
40. Harrower M., Brewer C. A. 2003 ColorBrewer.org: an online tool for selecting colour schemes for maps. *The Cartographic Journal* 40(1): 27-37.
41. Van der Walt S., Smith N. 2015. A better default colormap for Matplotlib SciPy 2015.
42. Van der Walt S. Smith N. 2020. MPL Colour Maps <https://bids.github.io/colormap>.
43. Crameri F. 2020. Scientific Colour Maps <http://www.fabiocrameri.ch/colourmaps>
44. Mikhailov A. 2019. Turbo, an improved rainbow colormap for visualization. *Google AI Blog*. <https://ai.googleblog.com/2019/08/turbo-improved-rainbow-colormap-for.html> accessed 05.06.2021
45. Reda K., Szafir D. A. 2021. Rainbows Revisited: Modeling Effective Colormap Design for Graphical Inference. *IEEE Transactions on Visualization and Computer Graphics* 27(2): 1032-1042.
46. Lautenschlager S., Figueirido B., Cashmore D. D., Bendel E. M., Stubbs T. L. 2020. Morphological convergence obscures functional diversity in sabre-toothed carnivores. *Proceedings of the Royal Society B* 287(1935): 20201818.
47. Rahman I. A., Lautenschlager S. 2017. Applications of three-dimensional box modeling to paleontological functional analysis. *Paleontological Society Papers* 22.
48. Morales-García N. M., Burgess T. D., Hill J. J., Gill P. G., Rayfield E. J. 2019. The use of extruded finite-element models as a novel alternative to tomography-based models: a case study using early mammal jaws. *Journal of the Royal Society Interface* 16(161): 20190674.

49. Lautenschlager S., Gill P. G., Luo Z. X., Fagan M. J., Rayfield E. J. 2018. The role of miniaturization in the evolution of the mammalian jaw and middle ear. *Nature* 561(7724): 533-537.
 50. Lautenschlager S., Witmer L. M., Altangerel P., Rayfield E. J. 2013. Edentulism beaks and biomechanical innovations in the evolution of theropod dinosaurs. *Proceedings of the National Academy of Sciences* 110(51): 20657-20662.
 51. Lautenschlager S., Witzmann F., Werneburg I. 2016. Palate anatomy and morphofunctional aspects of interpterygoid vacuities in temnospondyl cranial evolution. *The Science of Nature* 103(9-10): 79.
 52. Lautenschlager S. 2014. Morphological and functional diversity in therizinosaur claws and the implications for theropod claw evolution. *Proceedings of the Royal Society B: Biological Sciences*, 281(1785), 20140497.
 53. Mokrzycki W. S., Tatol M. 2011. Colour difference ΔE -A survey. *Machine Graphics and Vision* 20(4): 383-411.
 54. CIE Commission International de L'Eclairage. 1977. CIE recommendations on uniform color spaces color-difference equations and metric color terms. *Color Research & Application* 2 1 (1977): 5-6.
 55. Silva S., Santos B. S., Madeira J. 2011. Using color in visualization: A survey. *Computers Graphics* 35(2): 320-333.
 56. Kovesi P. 2015. Good colour maps: How to design them. *arXiv preprint arXiv:1509.03700*.
 57. R Core Team 2020. R: A language and environment for statistical computing. *R Foundation for Statistical Computing, Vienna, Austria*. <https://www.R-project.org/>.
 58. Caldwell B., Cooper M., Reid L. G., Vanderheiden G., Chisholm W., Slatin J. & White, J. 2008. Web content accessibility guidelines (WCAG) 2.0. *WWW Consortium (W3C)*, 290: 1-34.
 59. Schloss, K. B., Gramazio, C. C., Silverman, A. T., Parker, M. L., Wang, A. S. 2018. Mapping color to meaning in colormap data visualizations. *IEEE transactions on visualization and computer graphics* 25(1): 810-819.
 60. Sibrel, S. C., Rathore, R., Lessard, L., Schloss, K.B. 2020. The relation between color and spatial structure for interpreting colormap data visualizations. *Journal of Vision* 20(12): 7-7.
 61. Zeileis A., Fisher J. C., Hornik K., Ihaka R., McWhite C. D., Murrell P., Stauffer R., Wilke, C. O. 2019. colorspace: A toolbox for manipulating and assessing colors and palettes. *arXiv preprint arXiv 1903.06490*.
 62. Mayhorn C. B., Wogalter M. S., Bell J. L., Shaver E. F. 2004. What does code red mean?. *Ergonomics in Design* 12(4): 12-14.
 63. Dutel H., Gröning F., Sharp A.C., Watson P.J., Herrel A., Ross C.F., Jones M.E., Evans S.E., Fagan M.J., 2021. Comparative cranial biomechanics in two lizard species: impact of variation in cranial design. *Journal of Experimental Biology*.
 64. Ferrón H. G., Martínez-Pérez C., Rahman I. A., Selles de Lucas V., Botella H., Donoghue P. C. 2021. Functional assessment of morphological homoplasy in stem-gnathostomes. *Proceedings of the Royal Society B* 288(1943): 20202719.

Figures

Figure 1. Problems of the Rainbow colour scheme: (a) Non-uniform distances between individual colours (adapted from [27]). (b) Lack of intuitive perceptual order (c) Rainbow colour map as seen without and with colour vision deficiency (i.e. deuteranopia and protanopia type) and in greyscale.

Figure 2. Contour plots for different colour maps for Von Mises stress values shown for the simplified planar mandible model of the sabre-tooth cat *Dinofelis cristata*. In addition to the standard *Rainbow* colour map (A), nine further colour maps were tested: *Batlow* (B), *Cork* (C), *Inferno* (D), *Parula* (E), *Polar* (F), *Roma* (G), *Turbo* (H), *Viridis* (I), and *YIGnBu* (J). Grey regions in the contour plots represent stress magnitudes beyond the applied scale limit. R^2 -values are given for each colour map (see supplementary information for full correlation plots).

Figure 3. Contour plots for different colour maps for compressive and tensile stress values shown for the mandible model of the cynodont *Thrinaxodon liorhinus*. In addition to the standard *Rainbow* colour map (A), nine further colour maps were tested: *Batlow* (B), *Cork* (C), *Inferno* (D), *Parula* (E), *Polar* (F), *Roma* (G), *Turbo* (H), *Viridis* (I), and *YIGnBu* (J). Grey regions in the contour plots represent stress magnitudes beyond the applied scale limit. R^2 -values are given for each colour map (see supplementary information for full correlation plots).

Figure 4. Contour plots as seen without and with deuteranopia-type colour vision deficiency for different colour maps. Von Mises stress values shown for the cranium model of the dinosaur *Erlikosaurus andrewsi*. In addition to the standard *Rainbow* colour map (A), nine further colour maps were tested: *Batlow* (B), *Cork* (C), *Inferno* (D), *Parula* (E), *Polar* (F), *Roma* (G), *Turbo* (H), *Viridis* (I), and *YIGnBu* (J). Grey regions in the contour plots represent stress magnitudes beyond the applied scale limit. R^2 -values are given for each colour map (see supplementary information for full correlation plots).

Figure 5. Contour plots as seen without and with protanopia-type colour vision deficiency for different colour maps. Von Mises stress values shown for the cranium model of the capitosaurian temnospondyl *Parotosuchus helgolandicus*. In addition to the standard *Rainbow* colour map (A), nine further colour maps were tested: *Batlow* (B), *Cork* (C), *Inferno* (D), *Parula* (E), *Polar* (F), *Roma* (G), *Turbo* (H), *Viridis* (I), and *YIGnBu* (J). Grey regions in the contour plots represent stress magnitudes beyond the applied scale limit. R^2 -values are given for each colour map (see supplementary information for full correlation plots).

Figure 6 Contour plots as seen in full colour and grey scale for different colour maps. Von Mises stress values shown for a vertebra of the ornithischian dinosaur *Stegosaurus stenops*. In addition to the standard *Rainbow* colour map (A), nine further colour maps were tested: *Batlow* (B), *Cork* (C), *Inferno* (D), *Parula* (E), *Polar* (F), *Roma* (G), *Turbo* (H), *Viridis* (I), and *YIGnBu* (J). Grey regions in the contour plots represent stress magnitudes beyond the applied scale limit. R^2 -values are given for each colour map (see supplementary information for full correlation plots).

Figure 7. Contour plots depicted in the context of different background colours for the same model and colour maps. Von Mises stress values shown for a manual claw of the therizinosaurian dinosaur *Nothronychus graffami*. Grey regions in the contour plots represent stress magnitudes beyond the applied scale limit.

	Batlow	Cork	Inferno	Parula	Polar	Rainbow	Roma	Turbo	Viridis	YIGnBu
Von Mises	0.644	0.635	0.802	0.653	0.578	0.571	0.563	0.664	0.787	0.547
Tensile/compressive	0.910	0.871	0.905	0.959	0.934	0.887	0.872	0.852	0.933	0.967
Deuteranopia	0.890	0.696	0.693	0.460	0.604	0.504	0.476	0.117	0.475	0.511
Protanopia	0.547	0.555	0.738	0.642	0.458	0.564	0.485	0.717	0.876	0.670
Grey scale	0.476	0.440	0.578	0.360	0.348	0.086	0.098	0.008	0.306	0.500

Table 1. R^2 values for all tested colour maps, stress and visual appearances. Score with the highest value highlighted for each test setting.

Appendix B

Response to reviewers and list of requested changes to the manuscript

This paper has been reviewed by two researchers who are both enthusiastic about the results of the study. They both suggest only minor revision. I do not think it will take much time to incorporate the extra explanations that have been asked for, thus I recommend accept with minor revision.

S. Lautenschlager: I would like to thank the editor and the two reviewers for the constructive comments which have substantially improved the manuscript

Reviewer comments to Author:

Reviewer: 1

Comments to the Author(s)

This is an excellent study and very relevant now. The range of example models and colours is great to give an idea of what can work for future publications. The quantitative results are particularly helpful, because it gives some guidance on the best colour plot for the type of data, rather than a subjective preference based on appearance.

At the start, I expected to change my mind about the rainbow colour plot and want to use something different, but since you conclude that there is no single colour plot that solves “all” issues, I feel like the rainbow plot is still the one that will be chosen in future. Especially because it’s also the default plot from software and is easy to produce.

I particularly like the recommendation though to add a second set of contour plots (or more) with a different colour map in the supplementary material. This could really help those with CVD. I think adding multiple colour plots in the main article may be confusing, and most people are used to the rainbow colour plot.

S. Lautenschlager: Many thanks for the positive feedback.

One thing I’d like clearer is how to apply the colour plots; I found this confusing or unclear and if I was to do this myself, I wouldn’t know how. So, if this can be made clearer in the Methods or supplemental material, that would be great. I found the supplementary figures in the included files, but I could not find the supplementary material that was referred to on pages 4 and 6 in the Methods. If a step-by-step guide is included in the supplementary information, that would be good.

S. Lautenschlager: More information has been added to the methods (lines 82-88) and the used colour map codes (including instructions) have been added to the supplementary figures so that all supplementary information in one place.

Depending on the software, custom colour maps can be created. In this example, all colour maps were created in Abaqus via a command line script detailing the colour components via

HEX codes individually (see script in supplementary information). Alternatively, new colour maps (so-called spectra in Abaqus) can be created via a tools menu and selecting successive colours via a colour picker. This process will differ for individual software. However, specific pre-designed colour maps can be generated and accessed via online tools, such as Colourbrewer.org [40].

The thing I found most confusing is you say the outputs were generated in Abaqus and then converted using convertColor function in R? Were the outputs saved as images and converted from the default rainbow, or were the models exported and the colour was resampled in R? I would love to know how to do it.

S. Lautenschlager: *Outputs were generated using the specific colour maps directly in Abaqus and the view of the models was saved as image files. Colour information was then sampled from the images and converted using the R function (code is included in the supplementary information). More details have also been added to the methods (lines 80-82, 129-131)*

All colour maps used for this study consist of 24 individual colour values (definitions (order and HEX colour codes) are available in the supplementary information) and all outputs presented here were generated in Abaqus and model views were saved as image files.

...

The collection of the RGB colours from images of the FEA models and subsequent conversion to CIELAB colour space was done via the convertColor function in R [57] (see supplementary information).

Otherwise, very happy to recommend this for publication.

Reviewer: 2

Comments to the Author(s)

The manuscript has intriguing results for legible visualization that I hope to put into practice soon. Among minor improvements to the language, I suggest that the authors spell out acronyms more often, and be more specific for lines 217-218.

S. Lautenschlager: *As recommended by reviewer 2 all acronyms have been spelt out when first appearing under a new sub-heading. In addition, typos, grammatical aspects and minor comments highlighted in the PDF copy have been incorporated as suggested.*

More substantive but still minor improvements will be:

1. To explain more clearly the commercial propriety or openness of various color schemes, and how generally they're available amongst relevant simulation and visualization programs.

S. Lautenschlager: *More explanation has been added (as also requested by reviewer 1) (lines 71-73, 82-88).*

All colour maps are non-proprietary, in some cases versioned and available/defined via the respective references above. Not all colour maps are readily and equally available by default in all software but can be added in most cases (see also below).

...

Depending on the software, custom colour maps can be created. In this example, all colour maps were created in Abaqus via a command line script detailing the colour components via HEX codes individually (see script in supplementary information). Alternatively, new colour maps (so-called spectra in Abaqus) can be created via a tools menu and selecting successive colours via a colour picker. This process will differ for individual software. However, specific pre-designed colour maps can be generated and accessed via online tools, such as Colourbrewer.org [40].

2. To explain how well the mathematical discrimination between colors matches perceptual discrimination.

S. Lautenschlager: More detail has been added explaining the match between colour space and human perception (lines 218-224):

Human colour perception is not uniform, often subjective and dependent other factors such as age and individual variation and as such does not correspond to Euclidean distances in colour space [57]. The CIELAB colour space is an attempt to replicate human colour differentiation. As the correlation analysis only considers absolute changes along a trajectory, the analysis may not record the exact correlation when non-monotonic changes on the stress scale are associated with changes in different directions in the CIELAB space. However, this is less likely to be a problem for the perceptually-based colour maps.